

# Plasma antioxidants and oxidative stress status in obese women: correlation with cardiopulmonary response

Dyg Mastura Adenan[1], Zulkarnain Jaafar[2], Jaime Jacqueline Jayapalan[1] and Azlina Abdul Aziz[1]

[1] Department of Molecular Medicine, Faculty of Medicine, University of Malaya, Kuala Lumpur, Malaysia
[2] Department of Sports Medicine, University Malaya Medical Centre, Lembah Pantai, Kuala Lumpur, Malaysia

## ABSTRACT

**Introduction.** A high body fat coupled with low cardiopulmonary fitness and an increase in oxidative stress has been connoted as contributing factors in developing cardiovascular comorbidities. This study aimed to investigate the correlation between antioxidants and oxidative stress status with cardiopulmonary responses in women of different body mass index (BMI).

**Subjects and Methods.** Eighty female adults were recruited and divided into three groups; normal weight ($n = 23$), overweight ($n = 28$) and obese ($n = 29$), according to their BMI. Blood samples were obtained prior to cardiopulmonary exercise testing. Plasma samples were separated by centrifugation and analysed for enzymatic antioxidant activity including catalase, glutathione peroxidase and superoxide dismutase. Non-enzymatic antioxidant activities were assessed using 2, 2′-azino-bis (3-ethylbenzothiazoline-6-sulphonic acid) (ABTS) radical scavenging and ferric reducing ability of plasma (FRAP) assays. To evaluate the oxidative stress status of subjects, levels of reactive oxygen species and malondialdehyde, the by-product of lipid peroxidation, were measured. Cardiopulmonary responses were analysed using cardiopulmonary exercise testing (CPET) which involved 15 various parameters such as peak oxygen consumption, metabolic equivalents and respiratory exchange ratio.

**Results.** The obese group had significantly lower ABTS radical scavenging and FRAP activities than the normal weight group. A higher catalase activity was observed in the obese group than the normal weight group. Spearman's correlation showed an inverse relationship between catalase and peak oxygen consumption, while partial correlation analysis showed inverse correlations between superoxide dismutase and respiratory frequency, ABTS activity and oxygen pulse, and between ABTS activity and cardiac output.

**Conclusion.** Our results demonstrate a lower cardiovascular fitness and antioxidant capacity in obese women; the higher catalase activity may be a compensatory mechanism. The negative correlations found between these two parameters may indicate the potential effect of antioxidants on the cardiopulmonary system and deserve further analysis in a larger population. Nevertheless, this study provides the basis for future studies to further explore the relationships between redox status and cardiopulmonary responses. This can potentially be used to predict future risk of developing diseases associated with oxidative stress, especially pulmonary and cardiovascular diseases.

Corresponding author
Zulkarnain Jaafar,
zulkarnainj@um.edu.my

## INTRODUCTION

Obesity is considered as a 21st-century epidemic worldwide (*Rössner, 2002*). Its prevalence is higher in female adults as opposed to male adults, and several sociocultural factors have been associated with the disparities of weight gain between genders (*Kanter & Caballero, 2012*). Overweight and obesity are defined as exorbitant adipose tissue accumulation (*Rocha & Libby, 2009*) that could contribute to the pathogenesis of metabolic diseases, cardiovascular diseases, pulmonary diseases and certain types of cancer (*Singh et al., 2017*). The high incidence of overweight and obesity is associated with a sedentary lifestyle (*Ekkekakis et al., 2016*), in which low physical activity coupled with high consumption of energy-dense food over the needs of an individual would cause excess adipose tissue deposition in the body (*Doucet et al., 1998*).

High energy intake coupled with low energy expenditure causes the unused energy to be converted to and stored in the form of triglycerides in adipose tissues, either through an increase in adipocyte number (hyperplasia) or size (hypertrophy) (*Jung & Choi, 2014*). Accumulated adipose tissues in obesity trigger many reactions including the release of pro-inflammatory cytokines such as interleukin-6 (IL-6) and tumour necrosis factor alpha (TNF-$\alpha$) (*Kawasaki et al., 2012*). In addition, infiltration of immune cells into adipose tissues can lead to elevated generation of reactive oxygen species (ROS) (*Appari, Channon & McNeill, 2018*). At a low or moderate level, ROS act as secondary messengers in cell signalling cascades (*Kietzmann & Görlach, 2005*). However, an imbalance between the production of ROS and the ability of antioxidants to counteract these species, in favour of the former, can lead to oxidative stress (*Aruoma, 1998*). This can cause damage to important cellular structures such as membranes, proteins, lipids and nucleic acids (*Ceriello & Motz, 2004*). Increased oxidative stress in obesity is reported as one of the mechanisms for the pathogenesis of several chronic diseases such as cardiovascular diseases, diabetes and cancer (*Pizzino et al., 2017*).

In humans, redox status can be evaluated by measuring antioxidant activities and markers of oxidative stress. Antioxidant activities can be determined by measuring reducing activities, radical scavenging or chelating capacities of antioxidants. On the other hand, oxidative stress can be evaluated by measuring levels of ROS, activities of antioxidant enzymes as well as by-products of oxidative damage such as malondialdehyde and 8-hydroxy- 2′-deoxyguanosine (8-OHdG) (*Ho et al., 2013*).

Obese patients have a high level of fat mass (FM) and fat-free mass (FFM), which contribute to a higher circulating blood volume (*Van der Kooy, 1992*). High FM and FFM can augment the left ventricular stroke volume, and burden the heart, which may eventually lead to heart failure (*Collis et al., 2001*). Obese individuals may also have abnormal lung function as a result of increased weight, which in the long run, could lead to cardiovascular diseases such as heart failure and stroke (*Poirier et al., 2006*). Cardiopulmonary fitness that is assessed via exercise test provides an evaluation of overall exercise responses involving

pulmonary, cardiovascular and skeletal muscle system during exercise and at rest (*Albouaini et al., 2007*).

At present, there is still a lack of information on the association between antioxidants and oxidative stress status with cardiopulmonary fitness and body composition in humans, more so in females. Information on the association between redox status and cardiopulmonary fitness can be potentially used in predicting future risk of developing diseases associated with oxidative stress, especially pulmonary and cardiovascular diseases. Therefore, this study aimed to investigate the correlation between antioxidants and oxidative stress status with cardiopulmonary responses in women of different body mass index (BMI).

## MATERIALS & METHODS

### Study participants

All procedures involving human subjects complied with the Declaration of Helsinki 1975, as revised in 1983. The study protocol was approved by the Medical Research Ethics Committee of University Malaya Medical Centre (UMMC), Kuala Lumpur, Malaysia (reference number: 2017417-5141). Study participants were recruited among UMMC staff and students. Written informed consent was obtained from each volunteer. Female adults, aged 19–55 years old with any weight status except underweight were recruited. Exclusion criteria were subjects who smoke, subjects with any cardiovascular disease at any point of time, respiratory problems either restrictive or obstructive, musculoskeletal disabilities that could limit from performing exercise testing, and subjects with any medical conditions such as diabetes mellitus, epilepsy, hypertension, renal or thyroid problems.

### Anthropometric measurements and subjects' classification

For anthropometry measurements, we followed the anthropometry procedures manual guidelines of The National Health and Nutrition Examination Survey (NHANES). Subject's height was measured using Detecto ProDoc series physician scales (Detecto, China). Body weight and composition were measured using Bio-Impedance Analysis (InBody 270, USA) following the manufacturer's protocol. BMI is calculated based on the weight of an individual in kilogram divided by the square of height in meter (BMI $= kg/m^2$). The subjects were classified into three groups according to their BMI (*Lim et al., 2017*): Normal weight (18.5–22.9 $kg/m^2$), overweight (23–24.9 $kg/m^2$) and obese ($\geq 25$ $kg/m^2$).

### Sample collections

Blood sampling for oxidative stress and antioxidants markers were done prior to the CPET. The blood samples were collected from April to July 2018. Three millilitre blood was collected in K2EDTA tubes (BD, USA). The samples were centrifuged at 2,246 × g for 15 min at 4 °C. Plasma were separated, transferred to Eppendorf tubes and stored at −80 °C until further analyses.

## Biochemical analyses
### ABTS radical scavenging activity

2, 2′-azino-bis (3-ethylbenzothiazoline-6-sulphonic acid) (ABTS) radical scavenging activity in plasma was measured as previously described (*Re et al., 1999*). ABTS radicals were pre-generated by mixing 7 mM ABTS and 2.45 mM potassium persulfate in double-distilled water. The stock radical solution was incubated in the dark for approximately 16 h at room temperature. Absorbance of the stock solution was adjusted to 0.7 ± 0.02 at 734 nm and served as the working solution. Phosphate buffer saline (PBS, pH 7.4) was used to dilute the stock solution. Five μL of plasma was added to 200 μL of ABTS solution in a 96-well plate and incubated for 6 min and absorbance was taken at 734 nm using a spectrophotometer (Spectramax M3, USA). A standard curve was constructed using Trolox (0–2.4 mM). All analyses were performed in triplicate. The final results were expressed as Trolox equivalent antioxidant capacity (TEAC) value.

### Ferric reducing ability plasma (FRAP)

The ferric reducing ability of plasma was estimated as previously described (*Benzie & Strain, 1996*). FRAP reagent was prepared by mixing 25 mL acetate buffer (300 mM, pH 3.6), 2.5 mL 2, 4, 6-tripyridyl-s-triazine (TPTZ) solution (10 mM TPTZ in 40 mM HCl), and 2.5 mL $FeCl_3.6H_2O$ solution (20 mM). Briefly, 10 μL plasma was added to 40 μL double-distilled water. Then, 200 μL of FRAP reagent (warmed at 37 °C) was added to the sample and incubated for 8 min at 37 °C. The absorbance readings were taken at 600 nm against the blank. All analyses were performed in triplicate. The final results were expressed as μmol/L $FeSO_4$.

### Reactive oxygen species (ROS)

The fluorescence-based probe dichlorodihydrofluorescein diacetate (DCFH-DA) was used to detect for the presence of ROS (*Dikalov, Griendling & Harrison, 2007*). Briefly, 5 μL plasma and 100 μL of DCFH-DA reagent were added into a 96-black well plate. The mixture was incubated in the dark for 30 min at room temperature. The fluorescence readings were captured with excitation and emission wavelength of 485 nm and 530 nm, respectively on a fluorescence spectrophotometer (Spectramax M3, USA). All analyses were performed in triplicate. The final results were expressed as relative fluorescence unit (RFU).

### Lipid peroxidation activity

Thiobarbituric acid reactive substances (TBARS) assay was used to evaluate lipid peroxidation in the plasma samples (*Ramli et al., 2017*). Malondialdehyde (MDA), a by-product of lipid peroxidation in the sample will react with thiobarbituric acid to produce pink colored products. TBARS reagent was prepared by mixing 0.3 g TBA, 12 mL trichloroacetic acid and 1.04 mL of 70% perchloric acid in 80 mL double-distilled water. Fifty microliters of plasma samples and 250 μL of TBARS reagent were mixed together and heated for 30 min at 90 °C. Then, the mixture was left to cool on ice and subjected to centrifugation at 834 rpm at 25 °C for 10 min. Fifty microliters of the mixture were added into a 96-well plate and the absorbance reading was taken at 532 nm. 1, 1, 3,

3-tetraethoxypropane (TEP) was used as standard and analysed as above. All analyses were performed in triplicate. The final results were expressed as micromoles per litre ($\mu$mol/L).

### Catalase (CAT) activity

CAT activity was determined as previously described (*Góth, 1991*) with slight modifications (*Hadwan & Abed, 2016*). Ten microliters of plasma were incubated with 100 $\mu$L of substrate (65 $\mu$M hydrogen peroxide in 60 mM sodium-potassium phosphate buffer, pH 7.4) at 37 ° C for 3 min. The reaction was stopped by adding 400 $\mu$L ammonium molybdate (32.4 mM). The absorbance of the yellowish complex was determined at 397 nm. All analyses were performed in triplicate. The final results were expressed as enzyme unit (U).

### Glutathione peroxidase (GPx) activity

GPx enzyme activity was determined using a colorimetric assay kit (Cayman Chemical, USA) according to the manufacturer's protocol. GPx activity was estimated indirectly by a coupled reaction with glutathione reductase and NADPH. The oxidation of NADPH to NADP+ caused decrease in absorbance at 340 nm. The rate of decrease in absorbance at 340 nm is directly proportional to the GPx activity in the samples. Briefly, 100 $\mu$L of assay buffer, 50 $\mu$L of co-substrate mixture and 20 $\mu$L of sample were added into a 96-well plate. The reaction was initiated by adding 20 $\mu$L cumene hydroperoxide. The absorbance was read once every minute at 340 nm to obtain at least five time points. All analyses were performed in triplicate. The final results were expressed as nanomoles per minute per milliliter (nmol/min/ml).

### Superoxide dismutase (SOD) activity

SOD activity was determined using a colorimetric assay kit (Cayman Chemical, USA), according to the manufacturer's protocol. The SOD assay kit used tetrazolium salt for the detection of superoxide radicals generated by xanthine oxidase and hypoxanthine. Briefly, 200 $\mu$L diluted radical detector and 10 $\mu$L sample were added into the wells. The reaction was initiated by adding 20 $\mu$L diluted xanthine oxidase to the wells. The mixture was incubated for 30 min at room temperature and absorbance reading was taken at 450 nm. All analyses were performed in triplicate. The final results were expressed as unit per millilitre (U/ml) whereby one unit of SOD is defined as the amount of enzyme needed to exhibit 50% dismutation of the superoxide radicals.

### Cardiopulmonary exercise testing (CPET)

CPET was conducted at the human performance lab, Department of Sports Medicine, UMMC, Kuala Lumpur, Malaysia. Stationary cycle ergometer (Cosmed, Italy) was used, and the test was performed according to The Ramp protocol (*Takken et al., 2009*). The Ramp protocol involved a gradual increase of work rate in a stepwise pattern over time. Firstly, the subjects were required to warm-up on the unloaded pedalling for 1–3 min. The work rate was increased by 10 watts for each minute. Verbal encouragement was given to the subjects to exercise until they reached volitional exhaustion. On average, the test lasted for 8–12 min and the exercise testing provides useful physiological information including exercise work, metabolic response, metabolic gas exchange, ventilatory response,

cardiovascular response, and pulmonary gas exchange. Below were formulae of some of the parameters measured in CPET:

Peak oxygen consumption $=$ (Heart rate $\times$ Stroke volume) $\times$ [Arteriovenous oxygen difference]

Oxygen pulse $=$ Oxygen uptake $\div$ Heart rate

Metabolic equivalents $=$ [1.8(Work rate) $\div$ Body mass] $+3.5+3.5$.

## Statistical analyses

The sample size was calculated using G-Power version 3.1.9.2. The G-Power indicated that a minimum sample of 66 produced 95% confident level with effect size of $f = 0.40$, $\alpha = 0.05$ and 1-$\beta = 0.80$. Mean and standard deviation of maximum oxygen consumption (VO$_2$max) which is the indicator of fitness level were used based on study by *Pandey et al. (2014)*. The calculated sample size plus 20% dropout is 80 subjects. The Statistical Package for Social Sciences (SPSS) version 22.0 (IBM, NY, USA) was used to perform the statistical analyses. The Kolmogrov-Smirnov test was used to determine the distribution of the data. For variables that were normally distributed, one-way analysis of variances (ANOVA) was used to compare the mean differences of antioxidants and oxidative stress status and CPET parameters among the groups. The data were presented as mean $\pm$ standard deviation. In the post-hoc analyses, multiple comparisons of specific sample pairs were done. The Tukey post-hoc test was used when equal variances were assumed and Dunnett's T3 post-hoc test when equal variances were not assumed. The assumption was verified based on Levene's test of homogeneity. Correlation of the normally distributed variables were tested using Pearson's test and the partial correlation analysis was further performed to control for the effect of age on the redox and cardiopulmonary responses parameters. For skewed variables, Kruskal-Wallis test was used to compare the non-parametric variables among the groups. The non-normally distributed data were presented as median (25th, 75th percentiles). Spearman's test was used to determine the non-parametric relationship between the variables. A $p$-value less than 0.05 was considered significant.

## RESULTS

### Study subjects

The mean BMI for the study population was 25.94 $\pm$ 4.87 kg/m$^2$. Table 1 shows the descriptive characteristics of the study subjects. The subjects' height did not differ significantly among the three BMI groups: Normal weight (NW), overweight (OW) and obese (OB) ($p = 0.89$). The waist-hip ratio was significantly different between the OB-NW groups and OW-NW groups ($p < 0.001$). Percentage body fat, body fat mass and skeletal muscle mass were significantly different between the OB-NW groups, OB-OW groups and OW-NW groups ($p < 0.001$). The highest value in the aforementioned parameters was noted in the OB group.

### Biochemical analyses

The blood plasma was tested for antioxidant activities and oxidative stress status. The ABTS radical scavenging activity was significantly lower in OB [$F = 7.314$, $p = 0.001$, 95% CI

**Table 1  Characteristics of the study subjects.**

|  | Normal weight (n = 23) | Overweight (n = 28) | Obese (n = 29) | p value |
|---|---|---|---|---|
| Age (years)[d] | 25 (20,31) | 33 (25,42) | 35 (32,42) | <0.001[*] |
| Height (cm) | 155.35 ± 5.47 | 154.86 ± 6.29 | 154.60 ± 4.63 | 0.89 |
| Weight (kg) | 49.19 ± 4.60 | 60.46 ± 6.12 | 74.40 ± 9.45 | <0.001[*] |
| BMI (kg/m$^2$) | 20.40 ± 1.63 | 25.17 ± 1.32 | 31.07 ± 3.18 | <0.001[*] |
| WHR[d] | 0.81 (0.78, 0.85) | 0.88[c] (0.84, 0.90) | 0.91[a] (0.86, 0.96) | <0.001[*] |
| PBF (%) | 30.17 ± 5.23 | 37.91[c] ± 5.80 | 44.01[a,b] ± 4.75 | <0.001[*] |
| BFM (kg) | 14.93 ± 3.24 | 22.73[c] ± 4.51 | 32.99[a,b] ± 6.92 | <0.001[*] |
| SMM (kg) | 18.33 ± 2.19 | 20.37[c] ± 3.39 | 22.54[a,b] ± 2.64 | <0.001[*] |

Notes.

BMI, Body mass index; WHR, Waist-hip ratio; PBF, Percentage body fat; BFM, Body fat mass; SMM, Skeletal muscle mass.

[a] indicates significant difference between obese and normal weight groups.
[b] indicates significant difference between obese and overweight groups.
[c] indicates significant difference between overweight and normal weight groups.
[d] indicates data is presented as median (25th, 75th percentiles); Data for others are presented as mean ± standard deviation.
[*] indicates significant difference at $p < 0.05$.

$[-0.54, -0.02]]$ and OW [$F = 7.314$, $p = 0.001$, 95% CI $[-0.67, -0.15]]$ groups compared to the NW group (Fig. 1A). No significant difference was observed between the OB and OW groups ($p > 0.05$). FRAP activity was significantly lower in the OB group compared to NW [$F = 6.382$, $p = 0.003$, 95% CI $[-270.68, -17.99]]$ and OW [$F = 6.382$, $p = 0.003$, 95% CI $[-288.01, -46.23]]$ groups (Fig. 1B). The OW and NW groups did not differ significantly in the FRAP activity ($p > 0.05$). No significant differences in the ROS and MDA levels were observed among the different BMI groups ($p > 0.05$) (Figs. 1C and 1D). CAT activity was significantly higher in the OB [$F = 4.609$, $p = 0.013$, 95% $[0.70, 4.83]]$ and OW [$F = 4.609$, $p = 0.013$, 95% $[0.20, 4.06]]$ groups compared to the NW group (Fig. 1E). On the contrary, GPx and SOD activities did not differ significantly among the different BMI groups ($p > 0.05$) (Figs. 1F and 1G).

## CPET analyses

CPET analyses were done to determine the cardiac, pulmonary and respiratory gas exchange performances. Table 2 shows all the CPET analyses. Peak oxygen consumption (VO$_2$peak) was observed to be significantly lower in OB versus NW groups and OB versus OW groups ($p < 0.001$), however, no significant difference was seen between OW and NW groups ($p > 0.05$). A similar significant pattern was also observed in the metabolic equivalents (METs) parameter between OB versus NW groups and OB versus OW groups ($p < 0.001$). There was a decreasing trend in peak heart rate ($p = 0.04$) with increased BMI, but no significant difference was observed among the different BMI groups. On the other hand, oxygen pulse was found to be significantly higher in OB compared to NW groups and also in OW compared to NW groups ($p < 0.001$). The remaining parameters measured during CPET showed no significant differences among the different BMI groups of subjects ($p > 0.05$).

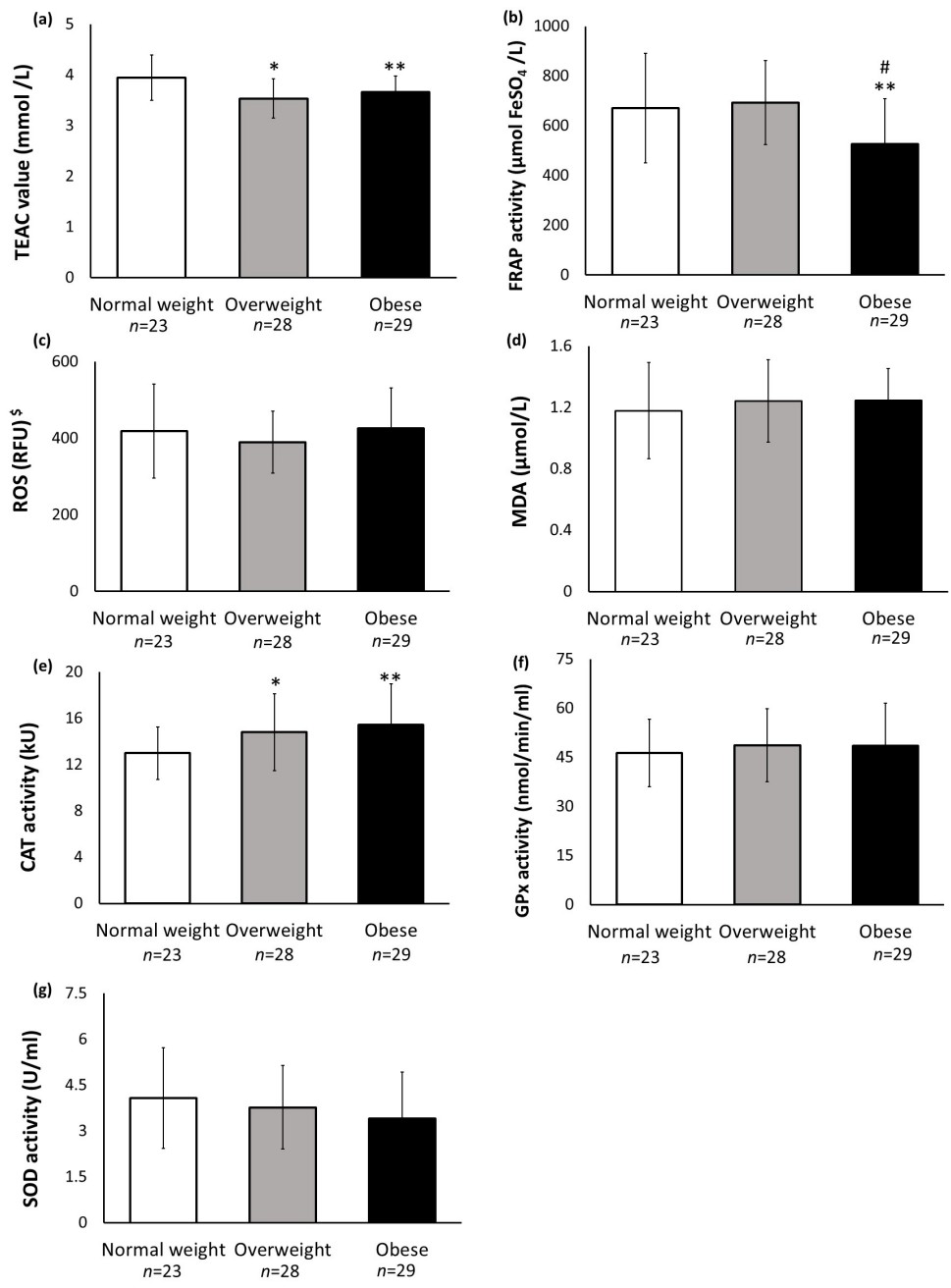

**Figure 1** **Biochemical analyses of plasma.** (A) ABTS radical scavenging activity, expressed as Trolox Equivalent Antioxidant Activity (TEAC); (B) Ferric reducing activity of plasma (FRAP) activity; (C) Reactive oxygen species (ROS) analysis, (D) Lipid peroxidation analysis, measured as malondialdehyde (MDA) level, (E) Catalase (CAT) activity; (F) Glutathione peroxidase (GPx) activity; (G) Superoxide dismutase (SOD) activity. $Data for ROS analysis is presented as median and error bar represents 25th, 75th percentiles; data for others are presented as mean and error bar represents standard deviation. * indicates significant difference between overweight and normal weight groups. ** indicates significant difference between obese and normal weight groups. # indicates significant difference between obese and overweight groups. The significant level was set at $p < 0.05$.

**Table 2 Cardiopulmonary exercise testing (CPET) analyses.**

| | Normal weight (n = 23) | Overweight (n = 28) | Obese (n = 29) | F or $x^2$ value | p value |
|---|---|---|---|---|---|
| Exercise duration (min) | 10.07 ± 2.27 | 9.13 ± 2.26 | 9.07 ± 1.30 | 1.95 | 0.15 |
| Peak oxygen consumption (ml/kg/min)[d] | 25.18 (23.39, 27.84) | 22.53 (20.01, 26.45) | 19.14[a,b] (17.25, 21.64) | 24.60 | <0.001[*] |
| Metabolic equivalents[d] | 6.90 (6.20, 8.00) | 6.35 (5.60, 7.60) | 5.30[a,b] (5.10, 5.90) | 22.76 | <0.001[*] |
| Respiratory exchange ratio | 1.26 ± 0.09 | 1.22 ± 0.10 | 1.22 ± 0.11 | 1.28 | 0.29 |
| Minute ventilation (L/min) | 46.96 ± 11.11 | 51.84 ± 11.61 | 50.97 ± 8.71 | 1.52 | 0.23 |
| Breathing rate (breath/min) | 57.04 ± 10.12 | 52.43 ± 8.15 | 52.69 ± 8.97 | 2.02 | 0.14 |
| Ventilatory tidal volume (ml/kg) | 1.01 ± 0.19 | 1.16 ± 0.22 | 1.14 ± 0.26 | 2.90 | 0.06 |
| Respiratory frequency | 46.37 ± 7.26 | 44.43 ± 7.65 | 46.18 ± 9.53 | 0.45 | 0.64 |
| Peak heart rate (beats/min) | 173.22 ± 9.94 | 166.04 ± 14.00 | 165.70 ± 8.71 | 3.48 | 0.04[*] |
| Oxygen pulse (mL/beat) | 7.08 ± 1.40 | 8.75[c] ± 1.72 | 8.53[a] ± 1.69 | 7.70 | <0.001[*] |
| Cardiac output | 7.66 ± 1.57 | 8.59 ± 1.45 | 8.53 ± 1.54 | 2.90 | 0.06 |
| Partial pressure of end-tidal carbon dioxide (mmHg) | 36.91 ± 4.03 | 37.11 ± 4.67 | 37.41 ± 3.95 | 0.09 | 0.91 |
| Partial pressure of end-tidal oxygen (mmHg) | 114.09 ± 5.11 | 114.14 ± 5.08 | 113.86 ± 5.05 | 0.02 | 0.98 |
| Ventilatory equivalent of carbon dioxide[d] | 31.20 (29.75, 34.15) | 31.15 (29.05, 34.30) | 31.20 (28.50, 35.20) | 0.05 | 0.98 |
| Ventilatory equivalent of oxygen[d] | 37.05 (35.60, 42.80) | 36.20 (33.05, 41.60) | 36.40 (32.90, 42.60) | 0.32 | 0.85 |

Notes.

[a]indicates significant difference between obese and normal weight groups.

[b]indicates significant difference between obese and overweight groups.

[c]indicates significant difference between overweight and normal weight groups.

[d]indicates data is presented as median (25th, 75th percentiles); Data for others are presented as mean ± standard deviation.

[*]indicates significant difference at $p < 0.05$.

## Correlation analyses between biochemical and CPET parameters

Data from the antioxidant activities and oxidative stress parameters, together with the CPET parameters, were analysed to determine their relationships. Table 3 shows the correlation analyses between the biochemical and CPET parameters. MDA was found to be inversely correlated with exercise duration ($r = -0.263$) while CAT activity demonstrated an inverse correlation with peak oxygen consumption ($r = -0.251$). ABTS radical scavenging activity was inversely correlated with both oxygen pulse ($r = -0.253$) and cardiac output ($r = -0.252$). The remaining parameters did not show any significant correlation ($p > 0.05$).

Due to the large age gap of the subjects, partial correlation analysis was also performed to control for the effect of age on the various parameters measured in the study. Table 4 shows the Pearson's partial correlation analyses between the biochemical and CPET parameters. Similar to Table 3, an inverse correlation was seen between ABTS radical scavenging activity with both oxygen pulse ($r = -0.26$) and cardiac output ($r = -0.25$). On the other hand, SOD activity showed an inverse correlation with respiratory frequency ($r = -0.23$), a correlation not previously observed. The remaining parameters did not demonstrate any significant differences ($p > 0.05$).

**Table 3 Correlation analyses between biochemical and CPET parameters.**

| | ABTS | FRAP | ROS[a] | MDA | CAT | GPx | SOD |
|---|---|---|---|---|---|---|---|
| Exercise duration | −0.146 | −0.195 | 0.141 | **−0.263**[*] | −0.237 | −0.008 | −0.043 |
| Peak oxygen Consumption[a] | 0.00 | 0.064 | −0.08 | −0.169 | **−0.251**[*] | 0.021 | 0.002 |
| **Metabolic equivalents**[a] | −0.036 | 0.109 | 0.031 | −0.185 | −0.157 | 0.015 | −0.01 |
| **Respiratory exchange ratio** | 0.126 | −0.008 | 0.144 | −0.027 | 0.006 | 0.151 | 0.046 |
| **Minute ventilation** | −0.126 | −0.062 | −0.014 | −0.056 | −0.029 | 0.083 | −0.091 |
| **Breathing rate** | 0.121 | 0.057 | 0.049 | −0.019 | 0.022 | −0.136 | 0.146 |
| **Ventilatory tidal volume** | 0.022 | −0.057 | −0.023 | −0.055 | −0.026 | −0.034 | 0.058 |
| **Respiratory frequency** | −0.139 | −0.057 | 0.033 | −0.023 | 0.034 | 0.165 | −0.187 |
| **Heart rate peak** | 0.057 | 0.07 | −0.065 | −0.023 | −0.049 | −0.095 | −0.098 |
| Oxygen pulse | **−0.253**[*] | −0.166 | −0.165 | −0.039 | 0.04 | 0.025 | −0.084 |
| Cardiac output | **−0.252**[*] | −0.132 | −0.065 | −0.099 | −0.078 | 0.04 | −0.125 |
| **Partial pressure end-tidal carbon dioxide** | −0.078 | −0.016 | 0.097 | −0.04 | −0.093 | −0.035 | −0.07 |
| **Partial pressure end-tidal oxygen** | 0.112 | 0.049 | −0.022 | 0.091 | 0.047 | 0.097 | 0.039 |
| **Ventilatory equivalent of carbon dioxide**[a] | 0.064 | 0.026 | −0.057 | 0.023 | 0.019 | 0.004 | 0.042 |
| **Ventilatory equivalent of oxygen**[a] | 0.084 | −0.039 | −0.009 | 0.041 | −0.028 | 0.048 | 0.028 |

Notes.

ABTS, ABTS radical scavenging activity; FRAP, Ferric reducing ability of plasma; ROS, Reactive oxygen species; MDA, Malondialdehyde; CAT, Catalase activity; GPx, Glutathione peroxidase activity; SOD, Superoxide dismutase activity.

[a]indicates non-distributed data and analysed using Spearman's correlation analyses; Other data is normally distributed and analysed using Pearson's correlation analyses; data is presented as correlation coefficients ($r$).

[*]indicates the correlation is significant at the 0.05 level (2-tailed).

**Table 4 Partial correlation analyses between biochemical and CPET parameters (age control).**

| | ABTS | FRAP | MDA | CAT | GPx | SOD |
|---|---|---|---|---|---|---|
| **Exercise duration** | −0.14 | −0.18 | −0.22 | −0.18 | 0.00 | −0.10 |
| **Respiratory exchange ratio** | 0.13 | 0.00 | −0.02 | 0.02 | 0.15 | 0.04 |
| **Minute ventilation** | −0.12 | −0.06 | −0.04 | −0.01 | 0.09 | −0.11 |
| **Breathing rate** | 0.13 | 0.06 | 0.00 | 0.05 | −0.13 | 0.13 |
| **Ventilatory tidal volume** | 0.02 | −0.07 | −0.09 | −0.06 | −0.04 | 0.09 |
| Respiratory frequency | −0.14 | −0.04 | 0.02 | 0.09 | 0.17 | **−0.23**[*] |
| **Heart rate peak** | 0.08 | 0.12 | 0.09 | 0.08 | −0.10 | −0.21 |
| Oxygen pulse | **−0.26**[*] | −0.18 | −0.07 | 0.00 | 0.02 | −0.06 |
| Cardiac output | **−0.25**[*] | −0.12 | −0.07 | −0.05 | 0.04 | −0.15 |
| **Partial pressure end-tidal carbon dioxide** | −0.08 | −0.01 | −0.03 | −0.09 | −0.03 | −0.08 |
| **Partial pressure end-tidal oxygen** | 0.11 | 0.05 | 0.08 | 0.04 | 0.10 | 0.05 |

Notes.

Partial correlation test was only done on normally-distributed data. Since peak oxygen consumption, metabolic equivalents, ventilatory equivalent of oxygen and carbon dioxide as well as reactive oxygen species were not normally distributed, these data are not included here. Data is presented as correlation coefficients ($r$).

ABTS, ABTS radical scavenging activity; FRAP, Ferric reducing ability of plasma; MDA, Malondialdehyde; CAT, Catalase activity; GPx, Glutathione peroxidase activity; SOD, Superoxide dismutase activity.

[*]indicates the correlation is significant at the 0.05 level (2-tailed).

## DISCUSSION

Obesity is a risk factor for developing cardiovascular diseases such as heart and coronary artery disease (*Yudkin et al., 2000*; *Van Gaal, Mertens & De Block, 2006*). At the meantime, oxidative stress and exercise capacity are the parameters that could be used to predict future cardiovascular risk in an individual (*Migliore et al., 2005*; *Myers et al., 2015*). Yet, the majority of studies have often analysed antioxidant activities, oxidative stress and cardiopulmonary responses separately and have not assessed their correlation. Thus, there is a lack of data on the actual relationship between these parameters. To overcome these limitations, our study analysed possible correlation between these parameters.

Antioxidants play important protective role against excessive production of free radicals, through scavenging or inhibiting their activities (*Nimse & Pal, 2015*). Plasma contains numerous low molecular weight antioxidants such as ascorbic acid, tocopherol, and uric acid (*Chevion et al., 1997*), which can be measured through several antioxidant assays. The ABTS assay measures the ability of antioxidants in plasma to scavenge ABTS radicals while the FRAP assay measures the capacity of antioxidants to reduce ferric ions (*Dasgupta & Klein, 2014*). In this study, the data shows that the ABTS scavenging activity and FRAP activity are lower in obese (OB) than the normal weight (NW) group. This implies the possibility of body fat mass influencing antioxidant status. Several studies have been conducted on the effects of body fat on antioxidants and oxidative stress status (*Savini et al., 2013*; *Lubrano et al., 2015*; *Lechuga-Sancho et al., 2018*). While majority of the studies demonstrated that high body fat was correlated with low antioxidant activities and high oxidative stress (*Amirkhizi et al., 2010*; *Hermsdorff et al., 2011*; *Jankovica et al., 2014*), some studies reported the opposite effect or no difference and these may be due to the dietary antioxidants intake of individuals (*Brown et al., 2012*; *Amaya-Villalva et al., 2015*; *Wang & Hai, 2015*; *Čolak et al., 2019*). However, results from our study seem to agree with previous reports on the inverse association of body fat with antioxidant status. The reduced antioxidant activities in the OB group could also be due to other factors such as dietary habits. A study had reported that women who followed a high quality diet, indicative of high intake of fruit and vegetables and low amounts of fat, were more likely to have lower BMI (*Boynton et al., 2008*). In addition, antioxidants may be mobilised elsewhere in the body to combat oxidative stress (*Clarkson & Thompson, 2000*; *Abdali, Samson & Grover, 2015*). However, some studies show that even with high dietary consumption of antioxidants-rich food, obese individuals still demonstrated deficiency of micronutrients (*Via, 2012*; *Soare et al., 2014*).

In addition to the low molecular weight antioxidant molecules, antioxidant enzymes are also protective against oxidative damage (*Jeeva et al., 2015*). The present data shows higher CAT activity in the OB group than the NW group. The low antioxidant activities and high CAT activity in the OB group are suggestive of the presence of oxidative stress, as reported by several publications (*Tinahones et al., 2008*; *Bausenwein et al., 2009*; *Zhang, Liu & Zong, 2016*). CAT is a tetrameric protein and is predominantly found in peroxisomes. It catalyses the conversion of hydrogen peroxide to hydrogen and water (*Ighodaro & Akinloy, 2018*). The increased CAT activity could be a compensatory mechanism to overcome the

increased hydrogen peroxide content. Our finding was in line with the study by Rindler who reported a significant increase in CAT activity in mouse fed with a high-fat diet, for 2–30 weeks (*Rindler et al., 2013*). Indeed, overexpression of CAT has been shown to be beneficial in preventing oxidative damage (*Awad, Aldosari & Abid, 2018*).

In this study, determination of oxidative damage was done by measuring levels of MDA and ROS in plasma. Results from the study demonstrate no significant differences in MDA and ROS levels among the different BMI groups. This implies that body fat does not contribute to the oxidative stress state. However, based on the findings of similar studies, the levels of MDA and ROS were reported to be significantly higher in the high BMI group (*Yesilbursa et al., 2004*; *Sankhla et al., 2012*; *An et al., 2018*). We speculated that the lack of significant differences in the oxidative stress parameters of the OB group, despite reduced antioxidant activities, could be due to the removal of the radicals by the increased CAT activity. It is also possible that changes in the levels of ROS and MDA in the OB group were too little that it could not be detected by the assays.

Disease development arises as a result of abnormal interactions among several systems. During an incremental exercise testing (CPET), the assessment of ventilation and volume of oxygen uptake and exhaled carbon dioxide can provide comprehensive data on several body systems simultaneously, namely cardiac, respiratory, haematological, and musculoskeletal systems. The present study shows several significant findings of CPET parameters among the different BMI groups. We found that $VO_2$peak was significantly reduced in the OB group; $VO_2$peak is the highest attainable oxygen uptake at which the work parameters plateau, and it allows the determination of aerobic exercise capacity of an individual. Similar to our study, several studies have also demonstrated lower $VO_2$peak in OB than NW subjects (*Miller et al., 2012*; *Shim et al., 2013*; *Bunsawat et al., 2017*). The reduced $VO_2$ peak point toward reduced exercise capacity, which may be caused by the gas exchange, cardiac, pulmonary, muscular, or effort limitations. A study conducted by *Green et al. (2018)* found that increasing BMI is associated with declined responses of $VO_2$peak and leg vascular conductance during cycling and isolated limb exercise. The reduction of $VO_2$ peak in the OB group may be caused by decreased vasodilation in contracting muscles during exercise. Also, changes in lipid profiles together with weight gain may also affect vascular dynamics during muscle contraction leading to these findings (*Steinberg et al., 1997*; *Clerk, Rattigan & Clark, 2002*; *Steinberg & Baron, 2002*).

Oxygen uptake increases as work increases, and our study shows that the cardiac output did not differ significantly among the BMI groups, and this may imply that cardiac output does not significantly contribute to the reduced $VO_2$peak in OB individuals in this study. The present study finds that MET was significantly lower in OB than NW subjects. MET relates to the individual's rate of oxygen uptake for a given work activity, and it dictates the functional capacity of an individual (*Ainsworth et al., 2011*). Oxygen is needed to fuel muscle contraction; thus, lower efficiency of energy expenditure during work resulted in poorer functional capacity. The lower MET in the OB group could be caused by the lower percentage of muscle mass (normalized to body weight) that is crucial for physical performance. Moreover, our study demonstrates a decreasing trend in the peak heart rate with increased BMI. This finding is supported by a study conducted by *Loftin et al. (2003)*
which also found that peak heart rate was lower in young OB female compared to the NW group. *Strandheim et al. (2015)* also reported that obesity and obesity-associated metabolic changes influenced both resting and peak heart rate during exercise. The deposition of adipose tissues, particularly in the cardiac muscles may hinder the cardiac function and exert unfavourable burdens to the heart (*Cole et al., 2011*). In addition, oxygen pulse, which represents the amount of oxygen consumed per heart rate ($VO_2/HR$) was higher in the OB and OW groups compared to the NW group. The increased oxygen pulse detected in OB and OW subjects in this study likely contributed to the increased cardiac output and stroke volume (*Datta, Normandin & ZuWallack, 2015*). *Laukkanen et al. (2006)* suggested that oxygen pulse value can be used to predict the risk of coronary heart disease.

With regards to the correlation between redox status and cardiopulmonary responses, our present study finds weak inverse correlations between CAT and $VO_2$peak. In response to the oxidative stress state, CAT activity would be increased to maintain the body's redox equilibrium, and likewise in this study we found that low exercise capacity level causes an elevation of the CAT activity in the OB subjects. Thus, the inverse correlation indicates a potential protective effect of exercise on prevention of lipid peroxidation. Interestingly, Lubkowska et al. in their study found that CAT activity could be reduced by engaging in regular moderate aerobic activity (*Lubkowska et al., 2015*).

After controlling the age, we found that SOD has weak inverse correlation with respiratory frequency, and ABTS activity has weak inverse correlations with oxygen pulse and cardiac output. Obesity has a profound effect on the respiratory physiology (*Parameswaran, Todd & Soth, 2006*) and it significantly affects respiratory function by decreasing lung volume, leading to increase in respiratory effort, oxygen consumption, and respiratory energy expenditure (*Littleton, 2011*; *Mafort et al., 2016*).  Higher respiratory frequency may further increase ROS production through electron transport chain reaction (*Salin et al., 2015*). Hence, SOD activity should be increased in order to compensate for the high ROS. In our study, we found a weak negative correlation between SOD activity and respiratory frequency, which means in high ROS state, the SOD activity was low. A few studies have reported that SOD activity and zinc level were markedly low among obese individuals (*Abdallah & Samman, 1993*; *Torkanlou et al., 2016*). Zinc is one of the cofactors of SOD, and it is possible that dietary zinc intake has an influence on SOD activity (*Olechnowicz et al., 2017*). In view of these, obese individuals may benefit from consuming food rich in zinc in order to increase their SOD activity and subsequently protect themselves against high ROS.

The increase in body mass in the OB individual leads to higher metabolic demand, increase oxygen requirements by the body, and increase adipose tissue perfusion, which subsequently leads to an increase in blood volume and augmentation of the cardiac output (*Cepeda-Lopez et al., 2019*). However, as a result of physical inactivity, the cardiovascular system develops an autonomic dysfunction (*Csige et al., 2018*; *Williams et al., 2019*) which leads to a decrease in the heart rate and heart rate peak, and as a compensatory mechanism, the cardiovascular system increases its stroke volume and oxygen pulse. Prolonged condition may lead to an enlargement of the heart called cardiomegaly (*Ebong et al., 2014*). Few studies have suggested that cardiomegaly is associated with a higher risk of

sudden cardiac death in obese subjects (*Abel, Litwin & Sweeney, 2008*; *Tavora et al., 2012*). The high oxygen consumption in OB individuals may lead to elevated rate of electron transport chain reaction, which subsequently produces a higher level of ROS in the body (*Salin et al., 2015*) meanwhile low ABTS radical scavenging activity may indicate the inability of antioxidant molecules to protect against oxidative stress in the OB group.

## CONCLUSIONS

In conclusion, obese women exhibited reduced cardiopulmonary fitness and antioxidant capacity, while catalase was increased, possibly as a compensatory mechanism. The negative correlations found between some antioxidants and cardiopulmonary response parameters indicate that antioxidants may affect the cardiopulmonary system and deserve further investigation, taking into consideration the effect of levels of physical activity, history of diet, supplement intake, the effect of hormones and duration of obesity. Nevertheless, this cross-sectional data provides the basis for future studies to further explore the relationships between redox status and cardiopulmonary responses parameters. This could predict future risk of developing diseases associated with oxidative stress, especially cardiopulmonary and cardiovascular diseases.

## ACKNOWLEDGEMENTS

We are grateful to all the subjects for their participation. The authors declare that there is no conflict of interest.

### Funding
This work was funded by the University of Malaya Research Fund Assistance (BK077-2017). The funders had no role in study design, data collection and analysis, decision to publish, or preparation of the manuscript.

### Grant Disclosures
The following grant information was disclosed by the authors:
University of Malaya Research Fund Assistance: BK077-2017.

### Competing Interests
The authors declare there are no competing interests.

### Author Contributions
- Dyg Mastura Adenan performed the experiments, analyzed the data, prepared figures and/or tables, authored or reviewed drafts of the paper, and approved the final draft.
- Zulkarnain Jaafar conceived and designed the experiments, performed the experiments, analyzed the data, authored or reviewed drafts of the paper, and approved the final draft.
- Jaime Jacqueline Jayapalan analyzed the data, authored or reviewed drafts of the paper, and approved the final draft.

- Azlina Abdul Aziz conceived and designed the experiments, authored or reviewed drafts of the paper, and approved the final draft.

## Human Ethics

The following information was supplied relating to ethical approvals (i.e., approving body and any reference numbers):

The Medical Research Ethics Committee of University Malaya Medical Centre (UMMC), Kuala Lumpur, Malaysia approved this research (reference number: 20174175141).

## Data Availability

Data are available as Supplemental File.

## Supplemental Information

Supplemental information for this article can be found online at http://dx.doi.org/10.7717/peerj.9230#supplemental-information.

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
