# Peer review of "Plasma antioxidants and oxidative stress status in obese women: correlation with cardiopulmonary response"

_PeerJ, doi:10.7717/peerj.9230_

## Round 0.1 · original submission · Major Revisions

Please consider all the revisions suggested by the two reviewers for your re-submission.

·

Basic reporting

1. Some language issues should be corrected, for example:
• Introduction, Line 2: “dynamics factors that have” should read “dynamic factors have”
• Introduction, Line 50 can be rephrased to clarify to “When energy intake is high and energy expenditure is low….1”
• Introduction, Line 53: “that presence” can be changed to read: “the occurrence of chronic inflammation present in adipose….”
• Introduction, Line 60-61 should be changed to: “Eventually, the elevated levels of ROS in a biological system could lead to an oxidative state.”
• Methods, Line 184: “as unit per milliliter” should be “units”
• Methods, Line 198: “Ramp protocol…” should be “The Ramp protocol”
• Methods, Line 203: “of antioxidant and oxidative stress status with CPET” should change to “of antioxidant and oxidative stress and CPET”
• Results, Line 228, 230, 233, 235, 245 the use of the words “decrease”, “change”, “increase” is not appropriate as it reflects a time component. It should rather be replaced with “lower”, “difference”, “higher” etc. Also check Discussion, Line 287, 296
• Discussion, Line 276-277: “Subsequently, lead to the elevated level of reactive oxygen species in the biological system result in oxidative stress state.” This sentence can be rephrased as follows: “These complications may include the elevation of reactive oxygen species and ultimately result in an oxidative stress state.”
• Some abbreviations were not explained upon first use, e.g. IL-6, TNF-α in the introduction and ABTS and FRAP in the abstract.

2. The introduction/background is sufficient to supply context, but some clarification is needed as indicated below. The references used are appropriate but some statements were not referenced and the reference style used in the reference list is not consistent.
• Sometimes the one idea does not flow to the next. Remember that each paragraph should be a unity describing an idea. For example in paragraph 2 the theme is mostly on inflammation, but at the end of that paragraph oxidative stress comes into play, while the whole third paragraph focus on oxidative stress. Also keep in mind that inflammation is not central to this article since it was not measured.
• In the introduction, the oxidative stress markers measured in the study was not introduced. When referring to previous studies in which these markers were measured, gives credence to your work.
• Sometimes it works better if the motivation moves on to the aim and not the other way around as it is currently stated in the last paragraph of the Introduction.
• In paragraph 2, Line 55 the following was stated: “An imbalance between pro-inflammatory and anti-inflammatory cytokines, in favour of the former, is thought to be the cause for the development of metabolic and cardiovascular disorders including atherogenesis.” Just keep in mind that multiple factors contribute the development of CVDs. Yes inflammation is a contributing factor, but not the only factor.
• The following statements needs references:
o Line 38: “Obesity is considered as a 21st-century epidemic worldwide.
o Line 44-46: “ …in which low physical activity coupled with high consumption of energy-dense food over the needs of an individual would cause excess adipose tissue deposition in the body.”
o Line 54-55: “…in which the enlarge adipocytes release pro-inflammatory cytokines such as IL-6 and TNF-α.”
o Line 60-61: “Eventually, the elevated levels of ROS in the biological system could lead to oxidative stress state.”
o Also check statements in Line 63-63, line 64-65, Line 70-71.
• In the reference list the following references lack page numbers or DOI numbers: Awad, Bunsawat, Cepeda-Lopez, Chan, Cole, Csige, Gustafson, Kawasaki, Kowalska, Limberaki, Lubkowska, Pizzino, Steinberg, Van Gaal

3. As indicated above, sometimes the one idea does not flow to the next. Please refer to examples given below:
• The first few lines (Line 268-279) of the discussion if mainly repetitive of the introduction. Also when the authors get to what they found some aspects of the results section is repeated, without immediately telling the reader why their findings is important.
• It may also help to structure the paragraphs similarly. Start off with what you found. How does it compare to known literature. If it is contradictory – why? What possible mechanisms may be at play?
4. The resolution of the figure were not good and the text is quite small.
5. The raw data was supplied

Experimental design

1. This article is original primary research and within the scope of the journal.
2. The research question was well defined, it is relevant and meaningful and it was stated how the research fills a gap in knowledge.
3. The investigation method complies to ethical standards but needs some clarification:
• The sample size was rather small, did you have enough power to address your hypothesis?
• Which guidelines were used for BMI classifications?
• It would have been interesting to see if the oxidative stress and antioxidant markers change after CPET
• For anthropometry, which guidelines and instruments were used?
• In the statistical analyses section it was stated that “All analyses were performed in triplicate” Does that refer to the biochemical analysis or the statistical analysis?
• Which of the variables did not display a normal distribution? It is not clear when Pearson vs Spearman correlations were used?

Validity of the findings

1. Limited information is available on the link between oxidative stress and cardiopulmonary fitness and body composition, especially in females and therefore this study is beneficial to fill this gap.
2. All underlying data have been provided but I do have some concerns:
• The difference in age of the groups, may be an bigger issue than indicated in the strengths and limitations of the study. Age, especially in women aged 19-55 may be a huge confounder when considering different hormonal levels in women in this age range. Do you have data on estradiol available? Or if the data of the main findings are normally distrusted, I would suggest adjusting for age in partial correlations (you may even consider data transformation).
• It is stated in the discussion in line 304 that CAT is more stable than SOD and GPx, please clarify?
• If a finding is not statistically significant, it should not be used to make an argument, please refer to lines 311 – 313 and line 372.
• The conclusion is a bit bold, and should be rephrased and toned down. This was cross-sectional work on a very small sample and does not show any predictive value.

Reviewer 2 ·

Basic reporting

English. The article needs editing. I include some suggestions (but not exhaustive editing) in comments to the authors.

References. Nothing to add

Structure and organization of the article. It has an acceptable format and the figures and tables re relevant.

Hypothesis. The article contains a hypothesis and results relevant to it.

Discussion. The results need to be better discussed and some paragraphs reorganized. At points the information given in the discussion is not directly related to their results.

Experimental design

Aims of the journal .The manuscript conforms to the aims and scope of the journal.

Scientific question. It is clearly indicated: it aims to fill the gap in knowledge on the relationship between oxidative status and cardiopulmonary fitness in relation to obesity.

Ethical and technical standards. The work conforms to the ethical standards and it has been conducted rigorously. I miss data on the diet and the level of physical fitness of the studied population and the fact that they have age differences. I appreciate that these aspects have been discussed as a limitation by the authors. I suggest that they can statistically re-analyze some of their results taking into account age and may be menopause (suggestions included in comments to the authors).

Methods. The methods have been described rigorously. I make some suggestions to the authors to include some details regarding th formula of the parameters used in the CPFT, which I think could improve the manuscript.

Validity of the findings

The manuscript adds the novelty to assess the possible relationship between oxidative status and cardiopulmonary function in obese women. Although the results do not allow for a global conclusion, since they did not assess the previous level of fitness (which may have an impact on the antioxidant status of the women), the data presented are relevant of interest to researchers in the field and provide ground for further studies.

Robustness of the data. The data are statistically sound. I think there are relevant data what can be easily included: I suggest to add the Skeletal Muscle Mass/body weight, Fat mass/body weight.

Conclusions. I think the authors have to be more cautious about the conclusions. The data indicate that in relation to obesity there is decrease in some antioxidants and an increase in catalase activity, likely due to a compensatory response. However, the assumption that catalase can predict metabolic disorders and complications, I think it is not supported by the data provided. There are many questions remaining (different ages of the population, level of fitness), which may have influenced. I think they should restrict to their original question and discuss what aspects remain in speculative field.

Additional comments

English. Editing by a native English speaker would improve the clarity of the manuscript and help the readier. Below I suggest a few modifications.
Line 25. Plasma samples were “separated”. I suggest to change to: were “obtained by centrifugation”
Line 39. Sociocultural dynamics factors that have been associated with…. I suggest to change to: “Sociocultural factors have been associated with…. “
Line 42:” lead to”. I suggest to change to: “contribute to”
Line 46: “What is more”. I suggest to delete this. This applies to similar sentences (i.e. in the abstract, in lines 70 and 346).
Line 67. The uncontrolled increase of oxidative “stress”. I think it is more correct to talk about increase in oxidative “damage”, since oxidative stress is just the disbalance between ROS production and elimination leading to damage to macromolecules.
Line 92. “Staffs” has to be changed to “Staff” (it is singular)
Line 111. Plasma layers were separated. I think you can delete “layers”
Line 187. “The exercise physiologist conducted CPET at the”. I suggest to change to: “CPET was conducted at the”
Line 325. “Ill interactions”. May be you can change to “abnormal interactions”.
Line 354. “lower in female youth”. I suggest to change to: “lower in young OB females”
Abbreviations. Some have to be included before in the text, such as NW, OB, OW, CPET

Abstract . It can be improved. I suggest to expand it with additional information on methods (i.e. what oxidative balance and cardiopulmonary parameters were assessed)

Methods. I suggest to include in methods if you asked about menopause, I also propose to include the formula of some of the parameters measured in the exercise testing, (i.e. the oxygen pulse; metabolic equivalents, peak oxygen consumption). I think this would help the reader.

Results. I think there are relevant data what can be easily included: I suggest to add the Skeletal Muscle Mass/body weight, Fat mass/body weight; You indicate that a drawback of the study is the fact that you found significant differences in age between the 3 groups. I suggest to re-analyze statistically the results taking into account age: 1) the relationship between age and the plasma parameters and 2) relationship between age and some of the cardiopulmonary parameters measured in CPET
Did you ask about menopause? If you can add some information it would improve the manuscript (either in the discussion if you did not ask about this, or if you know you can check if it influenced the results).

Discussion. I suggest to include the following aspects. !) to discuss on the type of antioxidants which are evaluated by FRAP and ABTS (low molecular weight antioxidants? Vitamins? ). This can be informative; 2) the possibility that you did not find differences in oxidative damage or ROS due to methodology (may be the method is not accurate enough to detect small differences, taking into account that ROS are highly unstable?
Lines 273-277. “Expansion of adipose tissues may activate inflammatory cascades that lead to secretion of adipocytokines, and together with changes in lipid and glucose metabolisms (Gustafon, 2010) may give rise to other complications when responses are uncontrolled. Subsequently, lead to the elevated level of reactive oxygen species in the biological system result in oxidative stress state”. I think this sentences need to be rephrased and organized according to the sequence of events: adipose tissue expansion, increase of pro-inflammatory adipokines, macrophage infiltration, ROS increase and tissue damage (also induced by alterations in lipid and glucose metabolism).

---

## Round 0.2 · Minor Revisions

Thanks you for the revisions made, which were well received by the reviewers. There are, some minor revisions that needs attention before the manuscript can be accepted. See attached comments from the reviewers.

·

Basic reporting

Thank you for your efforts in fixing the language errors. I did notice some minor issues that still needs ratification:
• Line 236: “The mean BMI for the study was” should be changed to “The mean BMI for the study population was”
• Line 302: “Yet, majority” should be changed to “Yet, the majority”
• Line 427: “OB individual” should be “OB individuals”

Experimental design

Thank you for your efforts in clarifying the mentioned issues on your experimental design. There are still some issues that I would like to bring to your attention:
• Please add the power calculation to manuscript.
• You indicated that you added a footnote to show the normal distribution when spearman vs Pearson correlations were used, but it was in fact not added.
• Line 37 and Line 288: You refer to Pearsons’ partial correlations. It should only be Partial correlations.”
• Line 278-279: It was stated: “Data from the antioxidant activities and oxidative stress parameters together with the CPET parameters were compared to determine the relationship between the various parameters.” This is not stated correctly. You did not compare these variables, but investigated associations/relationships/correlations between CPET parameters and markers of antioxidant activities and oxidative stress.

Validity of the findings

Thank you for your efforts in correcting the issues raised. Regarding the conclusion, you summarized the correlations found in the study, but you did not indicate what this mean? You should also conclude on that.

Reviewer 2 ·

Basic reporting

Adequate

Experimental design

Adequate

Validity of the findings

Adequate

Additional comments

The manuscript has been markedly improved and I think it deserves publication. I only suggest changing the conclusions, which I think are still a bit confusing. I suggest the modification of the following sentences:

Abstract: Lines 40-41. “Results from this study indicate potential correlation between redox status and cardiopulmonary response. However, validation of this cross-sectional data in a larger sample size and population would shed more light on the observed results”. I suggest to change to : “Our results indicate a lower cardiovascular fitness and antioxidant capacity in obese women; the higher catalase may be a compensatory mechanism. The correlations between antioxidants and cardiopulmonary responses deserve further analysis in a larger population”.

Conclusion. Lies 433-437. “In conclusion, female subjects who are obese showed higher CAT activity than normal weight subjects, despite the reduced non-enzymatic antioxidants activities and their metabolic responses were low, reflected by the levels of peak oxygen consumption and metabolic equivalents. The ABTS, CAT and SOD correlated negatively with oxygen pulse and cardiac output, peak oxygen consumption and respiratory frequency, respectively. Future studies should take…”. I suggest changing to: “Obese women exhibited reduced cardiopulmonary fitness and antioxidant capacity, while catalase was increased, possibly as compensatory mechanism. The negative relationship between some antioxidants and cardio-pulmonary responses to exercise, deserve further investigation, taking …

Minor comments:
Abstract line 44. I suggest deleting the word “parameters”
Line 317. The sentence “that even with excessive dietary consumption of antioxidants-rich food”. I suggest changing to: “that even with high dietary consumption of antioxidant-rich foods”.

---

## Round 0.3 · accepted · Accept

Thanks for addressing the minor revisions of reviewers.